# *Confidential-DPproof*: Confidential Proof of Differentially Private Training

**Ali Shahin Shamsabadi[1], Gefei Tan[2], Tudor Ioan Cebere[3], Aurélien Bellet[*4],**
**Hamed Haddadi[*1,5], Nicolas Papernot[*6,7], Xiao Wang[*2], Adrian Weller[*8,9]**
[1] Brave Software, [2] Northwestern University, [3] Inria, Univ Lille, [4] Inria, Univ Montpellier,
[5] Imperial College London, [6] Vector Institute, [7] University of Toronto, [8] University of Cambridge,
[9] The Alan Turing Institute

## Abstract

Post hoc privacy auditing techniques can be used to test the privacy guarantees of a model, but come with several limitations: (i) they can only establish lower bounds on the privacy loss, (ii) the intermediate model updates and some data must be shared with the auditor to get a better approximation of the privacy loss, and (iii) the auditor typically faces a steep computational cost to run a large number of attacks. In this paper, we propose to proactively generate a cryptographic certificate of privacy during training to forego such auditing limitations. We introduce *Confidential-DPproof*, a framework for Confidential Proof of Differentially Private Training, which enhances training with a certificate of the $(\varepsilon, \delta)$-DP guarantee achieved. To obtain this certificate without revealing information about the training data or model, we design a customized zero-knowledge proof protocol tailored to the requirements introduced by differentially private training, including random noise addition and privacy amplification by subsampling. In experiments on CIFAR-10, *Confidential-DPproof* trains a model achieving state-of-the-art $91\%$ test accuracy with a certified privacy guarantee of $(\varepsilon = 0.55, \delta = 10^{-5})$-DP in approximately 100 hours.

## 1 Introduction

Training Machine Learning (ML) models poses a potential threat to the privacy of users whose sensitive data is being analyzed. Indeed, ML models have been shown to leak information contained in individual training examples (Fredrikson et al., 2015; Shokri et al., 2017; Rahman et al., 2018; Song & Shmatikov, 2019; Balle et al., 2022; Carlini et al., 2022). To protect the privacy of training data, the de facto approach is to train models with guarantees of Differential Privacy (DP) (Dwork et al., 2014). This provides a statistical upper bound $\varepsilon$ on the privacy loss, i.e. the amount of information that a model leaks about individual training examples. Differentially Private Stochastic Gradient Descent (DP-SGD) is the canonical approach for training models under the framework of DP (Song et al., 2013; Abadi et al., 2016).

The guarantees of privacy provided by DP-SGD are derived analytically and constitute a theoretical upper bound $\varepsilon$ on the leakage of information from the training set once the DP-SGD algorithm was executed correctly. Unfortunately, it is difficult for an external party (i.e., *auditor*) to audit claims of privacy made by the modal trainer (i.e., *prover*) when they are only given query access to the model. Typically, in such black-box audits, the auditor must empirically measure the privacy loss of a trained model through the instantiation of practical attacks (Jagielski et al., 2020; Nasr et al., 2021; Lu et al., 2022; Zanella-Beguelin et al., 2023; Nasr et al., 2023). Such post hoc privacy auditing, however, suffers from some drawbacks: (1) it only establishes a lower bound on the privacy loss, as it is generally impossible to rule out the existence of better attacks that would yield a larger privacy loss; (2) computing tighter lower bounds compromises confidentiality, as decreasing the gap between the lower bound and the theoretical privacy guarantee $\varepsilon$ requires that the auditor be given more information about the training dataset and model (such as intermediate model updates, as shown in Figure 1 in Nasr et al., 2021); and (3) such improvements also come at a significant computational cost as the auditor needs to either train models millions of times (Nasr et al., 2021) or use many

---
[*]Alphabetical order

samples (Zanella-Beguelin et al., 2023) to obtain statistically reliable estimates of the privacy loss. Therefore, an auditor using these post hoc approaches can only hope to disprove a false claim of privacy as they cannot provide a certificate of the guarantees of DP-SGD privacy claimed by the prover when the audit is performed by a third party with limited access to the training phase.

In this work, we remediate these issues with the introduction of confidential proofs of DP training. We design *Confidential-DPproof* that enables a company to proactively prove to the auditor that their model was correctly trained using DP-SGD. This proof takes the form of a certificate which the auditor can use to verify the claimed value of $\varepsilon$. To obtain such a certificate, we rely on Zero Knowledge Proofs (ZKP) (Goldwasser et al., 1985; Goldreich et al., 1991). This allows us to prove statements about the training algorithm without revealing anything else about the training data or the model.

Our protocol for ZKP attests that the model trainer correctly executed the DP-SGD algorithm, i.e. that at each step i) it correctly subsampled a minibatch to benefit from privacy amplification by sub-sampling; ii) computed and clipped the associated per-example gradients to a preset norm to bound the influence of individual training examples on model updates; iii) added properly calibrated noise to the aggregated gradients; and iv) updated the model parameters using the noisy aggregated gradients. This is challenging for several reasons, the first of which is the existence of randomness at different stages of training performed with DP-SGD: this includes data subsampling to form minibatches and noise added to the gradients themselves. This randomness makes it possible for an adversarial prover (i.e., one that seeks to deviate from the DP-SGD training algorithm) to pick randomness that meets their needs. One avenue to address this problem would be for the prover to reveal its randomness to the auditor or let the auditor pick this randomness. However, this approach i) invalidates the guarantees of DP and thus risks the privacy of the prover as an informed adversary with knowledge of the randomness can infer information from the prover's training data; and ii) enables an adversarial auditor to bias the computations and blame the prover. To avoid this, we design a custom ZKP protocol that seeds the randomness privately and interactively so as to prevent its manipulation by provers while keeping it hidden from auditors: i) prior to the training, the prover commits to a random key $k$; ii) the auditor picks a fresh random value $r$ and sends it to the prover and iii) during the training, the prover proves in ZK that the unbiased randomness seed $s = k \oplus r$ is used without revealing $s$ or $k$ to the auditor.

In summary, we introduce desiderata for privacy auditing and propose the first ZK protocol which enables auditors to directly verify the exact privacy budget $\varepsilon$ of a DP-SGD training run, and thus provide a certificate of privacy. Because we rely on ZKPs, the prover does not need to reveal any information about their model and training dataset. *Confidential-DPproof* is an appealing alternative to the current status quo, where attacks are used to perform computationally costly audits. Our approach is made possible through the co-design of a DP training algorithm with a customized ZK protocol. We design and implement a specialized ZK proof protocol to efficiently perform the DP training. We implement and evaluate our framework in terms of accuracy, privacy, running times and communication costs. *Confidential-DPproof* affords low running time and high levels of accuracy.

## 2    PROBLEM STATEMENT

**Motivation.** Models that were trained without a privacy-preserving algorithm may leak information contained in their training set. For example, membership inference attacks (Shokri et al., 2017; Rahman et al., 2018) exploit access to a model's predictions to infer the presence (or absence) of a particular data point in its training set. It is also possible to reconstruct data points through an analysis of the model's parameters (Balle et al., 2022; Haim et al., 2022; Guo et al., 2022; Shamsabadi et al., 2023). Differential Privacy (DP) is the most established framework to reason about privacy leakage when analyzing data. It reasons about the properties of the algorithm itself rather than the data.

**Definition 1** (Differential Privacy (Dwork et al., 2014)). *A randomized algorithm $A$ is $(\varepsilon, \delta)$-DP if for any two neighbouring datasets $D$ and $D'$ and any measurable outcome $S \in Range(A)$:*

$$Pr[A(D) \in S] \leq e^\varepsilon Pr[A(D') \in S] + \delta. \tag{1}$$

Neighbouring datasets differ only in a single data point, and different variants for the definition of neighbouring datasets have been considered: adding-or-removing one record, zeroing-out one record or replacing one record (Ponomareva et al., 2023). We consider the *replace one* version, which is

most commonly used in machine learning and assumes that neighbourhing datasets $D$ and $D'$ have the same size. Therefore, the size of the dataset can be public, avoiding complications that might arise when designing our ZKP protocol.

Training with DP mitigates many attacks against the privacy of training data, including membership inference and training data reconstruction. Therefore, it is important for institutions that handle sensitive data to demonstrate that they train ML models under the framework of DP. These institutions must also be able to prove that their training run achieved a specific privacy budget $(\varepsilon, \delta)$: lower values of $\varepsilon$ imply a tighter bound on privacy leakage and thus a stronger privacy guarantee.

**Goal and notation.** We consider a setting with two parties: a prover and an auditor. The prover is an institution like a company or a hospital that wants to train a model on a sensitive dataset for different purposes such as providing ML-driven services. The auditor is an external entity that aims to verify the privacy guarantees of the prover's model. These two parties engage in communication to complete the audit. We identify the following desiderata for this audit:

- **Completeness.** The audit must infer the exact privacy guarantee and provide a certificate of this privacy guarantee. The guarantees of privacy provided by training with DP are derived analytically and constitute an exact upper bound $\varepsilon$, obtained based on the given privacy accounting technique, on the leakage of information from the training set. However, existing post hoc auditing approaches empirically estimate a lower bound of this leakage. There is a significant gap between the upper bound $\varepsilon$ that is claimed analytically (by the prover) and the empirical estimates of this upper bound that can be derived when the auditor uses existing post hoc privacy auditing methods with limited access to the training phase (Jagielski et al., 2020; Nasr et al., 2021; Lu et al., 2022; Zanella-Beguelin et al., 2023). Thus, it is difficult to draw conclusions as to whether the privacy claim made is valid or not. For instance, Nasr et al. (2023) obtain an audited value of $\varepsilon = 1.6$ while the theoretical upper bound is $\varepsilon = 8$.

- **Soundness.** The audit must be robust to malicious provers. Training models with DP often comes at a significant cost in utility (Tramer & Boneh, 2021; De et al., 2022). Furthermore, implementing DP training algorithms can be challenging and hard-to-detect errors can easily occur (Tramer et al., 2022) as a result of uncommon and complex modification to the underlying training optimizer (e.g., minibatch sampling, gradient clipping, noise calibration, and privacy accounting). Therefore, institutions may not adhere to their claims of training ML models with DP guarantees intentionally[1] (e.g., if they are trying to evade privacy regulations to maximize utility) or accidentally (e.g., if there is hard-to-detect bug in the implementation of their private algorithm).

- **Zero Knowledge.** The audit must preserve the confidentiality of the training data, model parameters, randomness used, and intermediate updates computed during training. Institutions are not willing to share their models and data with external parties due to privacy regulations and intellectual property concerns.

**Our solution.** We propose a zero-knowledge proof (ZKP) protocol to proactively generate a certificate of privacy during training while preserving the confidentiality of training data, intermediate model updates and trained model. An auditor can then use this certificate to confirm that the prover did train their model on a private dataset with a specific DP guarantee. A ZKP protocol $\Pi$ allows a party to prove to an auditor that the evaluation of an agreed upon circuit $\mathcal{C}$ representing a program on a private input *Inp* is *Out*, i.e., $\mathcal{C}(Inp) = Out$, while revealing no additional information about *Inp* (Goldwasser et al., 1985; Goldreich et al., 1991). In our paper, the private input *Inp* of the prover consists of the training dataset $\mathcal{D}$ and the random seed $s$. The circuit $\mathcal{C}$ represents a privacy-preserving training algorithm which the prover and auditor agreed on, while *Out* is the correct output of that algorithm. In the following, we consider the DP-SGD algorithm (Abadi et al., 2016), the prevailing approach to differentially private training.[2] The circuit $\mathcal{C}$ also captures specific values for the hyperparameters of DP-SGD – such as its noise multiplier, clipping norm, minibatch size and number of training

---

[1]For example see Apple's ambiguity around not releasing the exact privacy budget of a deployed DP mechanism and/or other relevant details (Tang et al., 2017). More generally, institutions often have incentives to not train models under the framework of DP or consider a large privacy budget to get a better utility.

[2]Our approach can be easily extended to other DP training algorithms. We illustrate this on the DP-FTRL algorithm (Kairouz et al., 2021) in Appendix D.

---

**Algorithm 1:** Differentially Private Stochastic Gradient Descent (DP-SGD)

---

**Input:** Training dataset $\mathcal{D} = \{X, Y\}$, clipping norm $C$, learning rate $\eta$, noise multiplier $\sigma$, loss function $\mathcal{L}(\cdot)$
**Output:** Parameters

---

1: Initialize $W$ randomly
2: **for** $t \in T$ **do**
3:      Sample a minibatch $\{X^{(t)}, Y^{(t)}\} \sim \{X, Y\}$ randomly with probability $l/n$
4:      **for** $\{\mathbf{x}_i, y_i\} \in \{X^{(t)}, Y^{(t)}\}$ **do**
5:          $\mathbf{g}_i = \nabla_{W_t} \mathcal{L}(W_t, \{\mathbf{x}_i, y_i\})$              ▷ Compute per-example gradient
6:          $\mathbf{g}_i \leftarrow \mathbf{g}_i \cdot \min(1, \frac{C}{\|\mathbf{g}_i\|_2})$             ▷ Clip per-example gradient
7:      $\mathbf{g} \leftarrow \frac{1}{l}(\sum_{i=1}^{l} \mathbf{g}_i + \mathcal{N}(0, \sigma^2 C^2 \mathbf{I}))$        ▷ Add calibrated noise
8:      $W^{t+1} \leftarrow W^t - \eta \mathbf{g}$               ▷ Update model parameters
9: **return** $W^T$                      ▷ Final model parameters

---

iterations. These hyperparameters influence the strength of the $(\varepsilon, \delta)$-DP guarantee. Put altogether, our ZK proof protocol $\Pi$ has the above desired properties: *Completeness* – For any input training data and a model that is trained on this data with the correct $(\varepsilon, \delta)$-DP guarantees in the clear, an honest prover (who behaves correctly) can convince an honest auditor that $\mathcal{C}(Inp) = Out$ using $\Pi$;[3] *Soundness* – Given an input training data and a model that is trained on this data with incorrect $(\varepsilon, \delta)$-DP guarantees in the clear, no malicious prover (with the ability to behave arbitrarily) can falsely convince an honest auditor that $\mathcal{C}(Inp) = Out$ using $\Pi$; and *Zero Knowledge* – If the prover and auditor execute $\Pi$ to prove that $\mathcal{C}(Inp) = Out$, even a malicious auditor (with the ability to behave arbitrarily) learns no information about the training data, intermediate updates and trained model other than what can be inferred from the fact that the model is trained with the DP-SGD algorithm with $(\varepsilon, \delta)$-DP guarantees on the training data.

## 3   *Confidential-DPproof*

We design a framework, *Confidential-DPproof*, that confidentially and efficiently certifies the upper bound on the privacy leakage $\varepsilon$ achieved by a training run of DP-SGD, foregoing the need to empirically audit the protections afforded by the resulting model. *Confidential-DPproof* works as follows:

- Prover publicly announces the privacy guarantees $(\varepsilon, \delta)$ that it is planning to achieve.

- Prover and auditor agree on the DP-SGD training algorithm and specific values for its hyperparameters including minibatch size, noise multiplier, clipping norm, learning rate, number of iterations and loss function. These hyperparameters are set to achieve the claimed privacy guarantees $(\varepsilon, \delta)$ while retaining the ability to learn models with high utility.

- Prover and auditor run our *interactive* zero-knowledge protocol and provide a certificate for the claimed privacy guarantee $(\varepsilon, \delta)$. We represent the whole procedure as one single public circuit, allowing the prover to generate one single proof demonstrating to the auditor that it correctly executed all of the steps of the DP-SGD algorithm (see Algorithm 1).

Our zero-knowledge proof protocol can be decomposed into three main phases: i) data commitment; ii) interactive and private randomness seed generation; and iii) zero-knowledge proof of the DP-SGD training. Below, we describe each phase of *Confidential-DPproof* in detail.

**Phase 1: Data commitment.** The prover commits to the training data $\mathcal{D}$. Let $\mathcal{D} = \{X, Y\}$, be the privacy-sensitive dataset owned by the prover in which $X = \{\mathbf{x}_1, ..., \mathbf{x}_n\}$ and $Y = \{y_1, ..., y_n\}$ denote data points and ground-truth labels, respectively, collected from users. These commitments are binding and hiding, meaning that the prover cannot change the content being committed without the auditor's consent and that the commitment does not reveal anything about the underlying content.

We use Information-Theoretic Message Authentication Codes (IT-MACs) as the basis of this data commitment (Franzese et al., 2021). For each bit $x$ possessed by the prover, the prover holds $(x, M)$ and the auditor holds $K$ such that $K = M \oplus x\Delta$ where $\Delta$ is a global authentication key uniformly generated by the auditor, and $\oplus$ denotes XOR. This algebraic relationship prevents the prover, at each

---

[3]The $(\varepsilon, \delta)$-DP guarantees are obtained by relying on a (public) privacy accounting technique. Ensuring the correctness of privacy accounting and of its underlying theoretical derivations are out of our scope.

---

**Algorithm 2:** Unbiased randomness seed commitment

---

**Input:** Commitment scheme {Commit, Reveal}, Hash function $H(\cdot)$, digital signature scheme {Gen, Sign, Verify}
**Output:** Random seed $s$ and signature $v$

1: Prover commits to a uniformly random value $k$, obtaining $[\![k]\!] := $ Commit $(k)$, and send $[\![k]\!]$ to the auditor.
2: Auditor runs Gen($1^n$) to get signing key sk and verification key pk, generates a uniformly random value $r$, computes
   $v = $ Sign$_{\mathsf{sk}}(H([\![k]\!], r))$, and send $v$, $r$, pk to the prover.
3: Prover sets the seed as $s = k \oplus r$

---

step of the computation, from modifying $x$ in any ways that the auditor does not agree to. For more details, please see Appendix B.

**Phase 2: Interactive and private randomness seed generation.** The privacy guarantees of DP-SGD stem from the introduction of randomness at different stages of training – randomness which is calibrated to the sensitivity of the training run to individual training points. Two randomization primitives contribute to privacy in DP-SGD: data subsampling and the addition of noise to the clipped gradients. However, an adversarial prover can bias these sources of randomness to increase the accuracy of their model at the expense of privacy, for instance by repeatedly running the training algorithm and selecting the seed that yields the best accuracy, or by deliberately choosing noise terms that have small magnitudes. In this case, the prover should not be able to demonstrate that the training run provides $(\varepsilon, \delta)$-DP guarantees. To ensure that the prover cannot perform such manipulations, we design an unbiased randomness commitment protocol (Algorithm 2) that interactively generates an unbiased randomness seed, and then commits that seed prior to beginning training. First, the prover commits to a random value $k$ and sends the commitment $[\![k]\!]$ to the auditor. Next, the auditor generates a random $r$, computes and signs the sha256 hash $v = $ Sign$_{\mathsf{sk}}(H([\![k]\!], r))$ where sk is the signing key, and sends both the signature $v$ and the random value $r$ to the prover. The prover then sets the unbiased randomness seed as $s = k \oplus r$, and uses it to derive all the randomness in the algorithm. Note that the unbiased randomness seed $s$ is random and cannot be biased by the prover or auditor, which guarantees an unbiased source of randomness when subsampling data or noising clipped gradients. This is because the prover and the auditor cannot see values generated by other parties which contribute to the computation of $s$. When executing DP-SGD, the prover demonstrates that it used the correct seed $s$ by verifying the signature $v$ in Algorithm 3. We use SHA-256 as the hash function modeled as random oracle (Bellare & Rogaway, 1993) and AES-128 as the pseudorandom generator. The hiding and binding quality follows from the security of the underlying scheme for obtaining the commitment and digital signature. We refer to Goldreich (2004) for more detailed discussion on commitment schemes and digital signatures.

**Phase 3: Proving the privacy guarantees in ZK.** The prover must follow the DP-SGD training algorithm exactly and prove that i) $l$ data points were sampled randomly to form a minibatch $X^{(t)}$ at each iteration $t$; ii) the associated gradients of a loss function $\mathcal{L}(\cdot, \cdot)$ with respect to the model parameters $W_t$ are computed on a per-example basis, $\mathbf{g}_i = \nabla_{W_t} \mathcal{L}(W_t, \mathbf{x}_i) \forall \mathbf{x}_i \in X^{(t)}$ so that no information is shared across individual data points; iii) the contribution of each example to the gradient are bounded by clipping the $l_2$ norm of these individual per-example gradients by $C$, $\mathbf{g}_i \leftarrow \mathbf{g}_i \cdot \min(1, \frac{C}{\|\mathbf{g}_i\|_2})$; iv) draw $m$ (cardinality of each per-example gradient) independent noise samples $\mathbf{n} = [n_1, .., n_m]$ from a Normal distribution scaled by $C$ and the noise multiplier $\sigma$, $n_1, ..., n_m \sim \mathcal{N}(0, \sigma^2 C^2)$, and add $\mathbf{n}$ to the sum of all clipped per-example gradients $\mathbf{g} \leftarrow \frac{1}{l}(\sum_{i=1}^{l} \mathbf{g}_i + \mathbf{n})$; v) model parameters are updated with the aggregated noisy gradients once multiplied by the learning rate $\eta$, $W^{(t+1)} \leftarrow W^{(t)} - \eta \mathbf{g}$; and vi) a total of exactly $T$ training iterations were performed. Then, the auditor can verify the claimed privacy guarantee based on the fact that the DP-SGD training algorithm was executed correctly *and* executed with the specific hyperparameter values for the minibatch size, number of iterations, and noise multiplier.

We represent all the above-mentioned computations in circuits and use EMP (Wang et al., 2016) to encode all operations faithfully and our ZK protocol proves that the circuit evaluation is done correctly. Next, we describe our ZK design (Algorithm 3) for steps involving randomness.

Regarding data subsampling in step (1), we shuffle data points based on a random value generated through Algorithm 2 and then partition the shuffled array of data points into minibatches of the predefined size – starting from the first data point in the shuffled array. This "random shuffle" strategy is often used in practice even though it does not completely match the privacy accounting of

---

**Algorithm 3:** Zero Knowledge Proof of DP-SGD Training

---

**Input:** Input dataset $\mathcal{D} = \{X, Y\}$, final model parameters $W^T$, Zero Knowledge Proof System $\{\mathcal{P}, \mathcal{V}\}$
**Output:** Accept or Reject

1: Prover commits to the dataset $\mathcal{D}$
2: Prover and Auditor run interactive Algorithm 2 to obtain $s$ and $v$, and prover commits to $s$
3: Prover and Auditor agree on circuit $\mathcal{C}_{C,\eta,\sigma,\mathcal{L}(\cdot)}$ representing the public Algorithm 1 with fixed input hyperparameters $\{C, \eta, \sigma, \mathcal{L}(\cdot)\}$, the circuit takes input $s$ as randomness seed and $\mathcal{D}$ as input training dataset.
4: Prover runs the algorithm $\mathcal{P}$ to generate a proof $\pi$ for the statement: $\mathcal{C}_{C,\eta,\sigma,\mathcal{L}(\cdot)}(\mathcal{D}, s) = W^T$ **and** $s = k \oplus r$ **and** $\mathsf{Verify}_{\mathsf{pk}}((H(\llbracket k \rrbracket), r), v) = 1$
5: Auditor runs the algorithm $\mathcal{V}$ on the proof $\pi$, and output **Accept** if and only if $\mathcal{V}$ accepts; else outputs **Reject**

---

DP-SGD (Tramer & Boneh, 2021). We note that our unbiased randomness commitment protocol can be easily extended to support Poisson sampling (as considered by the privacy accounting of DP-SGD) with negligible cost: each training sample is selected by independently flipping a biased coin with probability $l/n$ based on a random value generated through our unbiased uniform randomness.

When it comes to sampling independently drawn normally distributed noise to perturb each dimension of aggregated gradients in step (iv), efficiency is paramount. Thus, we first generate uniformly distributed random values and then leverages the Box–Muller transform Box & Muller (1958) to convert them to the desired normally distributed noise. In particular, our confidential noise sampling protocol works as follows: i) Fix the randomness seed by running Algorithm 2; ii) Generate two uniformly distributed random values $U_1$ and $U_2$ in ZK using a pseudorandom generator with the above-computed seed $s$; iii) Compute in ZK $z_1 = \sqrt{-2 \ln u_1} \cos(2\pi u_2)$ and $z_2 = \sqrt{-2 \ln u_1} \sin(2\pi u_2)$ (these two values are i.i.d. following normal distribution ensured by Theorem 1); and iv) Create $n_1 = \sigma C z_1$ and $n_2 = \sigma C z_2$ that are distributed according to a scaled normal distribution $\mathcal{N}(0, \sigma^2 C^2)$.

**Theorem 1.** *If $u_1$ and $u_2$ are two independent and uniformly distributed random values, then $z_1 = \sqrt{-2 \ln u_1} \cos(2\pi u_2)$ and $z_2 = \sqrt{-2 \ln u_1} \sin(2\pi u_2)$ are independent random values from a normal distribution with mean zero and unit variance.*

*Proof.* Computing the distribution of $\sqrt{-2 \ln u_1}$ and using the transformation law (Theorem 1.101 (Klenke, 2013)), we get the following joint density of $z_1$ and $z_2$:

$$f(z_1, z_2) = \frac{-1}{\sqrt{2\pi}} e^{\frac{-z_1^2}{2}} \frac{-1}{\sqrt{2\pi}} e^{\frac{-z_2^2}{2}} = f(z_1)f(z_2), \tag{2}$$

demonstrating that $z_1 \& z_2$ are independent and normally distributed with mean 0 and variance 1. $\square$

## 4 EXPERIMENTAL EVALUATION

Since we design *Confidential-DPproof* to enable institutions to prove the DP guarantees of their model to an auditor while protecting the confidentiality of their data and model, our experiments consider the following major dimensions: i) What are the best trade-offs between utility and certified privacy guarantees that *Confidential-DPproof* can achieve given the respective interests of the institution training a model (in terms of utility) and the auditor (in terms of privacy)? ii) What is the cost of verifying the training run while ensuring confidentiality? In particular, how does this cost scale as a function of the number of training iterations? To answer these questions, we evaluate the performance of *Confidential-DPproof* as follows: i) **Effectiveness:** We design an empirical evaluation of our DP training algorithm in achieving high accuracy in a limited number of iterations; ii) **Efficiency:** We implement our customized ZKP protocol in EMP-toolkit (Wang et al., 2016) and evaluate its running time in proving the DP guarantees.

**Dataset and Feature extractors.** We consider two common datasets (see Appendix A), for DP-SGD training benchmarking Tramer & Boneh (2021); Papernot et al. (2021); De et al. (2022); Shamsabadi & Papernot (2023): CIFAR-10 and MNIST. Verifying the entire training procedure of complex models calls for heavy cryptographic machinery and the associated computational cost would be prohibitively expensive in most settings of interest once it is implemented cryptographically. We address this issue by incorporating feature extractors (Tramer & Boneh, 2021) to enable DP-SGD training of simpler models. We still achieve accuracy-privacy tradeoffs that are competitive with the SOTA: instead of training a complex model on a raw dataset with DP-SGD, we train a logistic

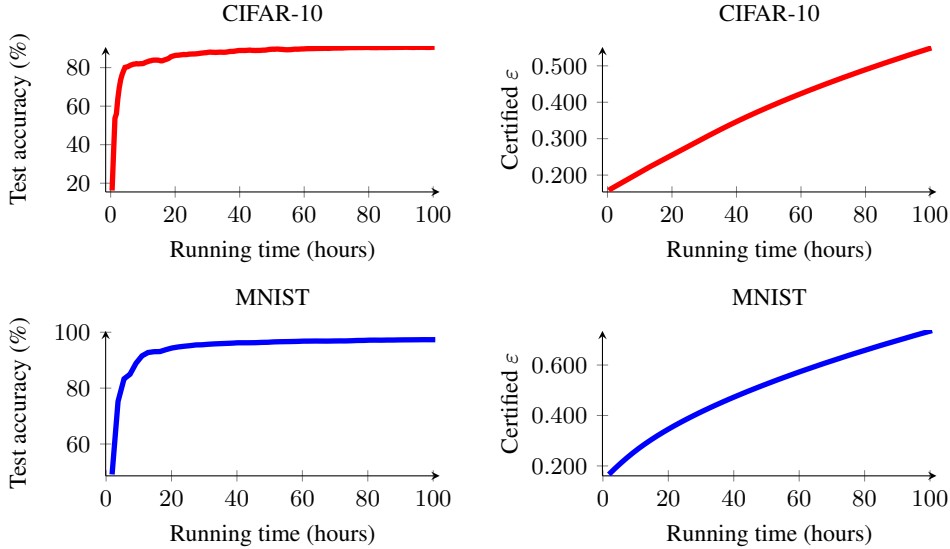

Figure 1: Accuracy and privacy guarantees achieved by *Confidential-DPproof* as a function of its running time. The running time is computed based on the number of DP training iterations for various DP budgets ($\varepsilon, \delta = 10^{-5}$). A prover can obtain its desired privacy certificate by proving that they correctly executed DP-SGD algorithm over a sufficient amount of time using our ZKP protocol without revealing any information about its training data and model. ***Confidential-DPproof* trains models achieving high test accuracy with strong certified privacy guarantees in practical running time.**

regression on an embedding of the data. In particular, we extract features for CIFAR-10 and MNIST from the penultimate layer of a SimCLR model (Chen et al., 2020) and ScatterNet (Lowe, 1999) (see Table 2), respectively. Following (Tramer & Boneh, 2021), we use unlabeled ImageNet, which is considered as public data (Tramèr et al., 2022) coming from a different distribution compared to the distribution of the private data, for training SimCLR-based feature extractor. ScatterNet is a non-learnable SIFT-like feature extractor that uses a cascade of data-independent wavelet transforms. We then train a logistic regression model with the DP-SGD algorithm on these extracted features. In a private setting, training logistic regressions on extracted features outperforms training complex networks in terms of accuracy-privacy tradeoff (Tramer & Boneh, 2021). We also demonstrate that it significantly improves the efficiency of confidential proof of DP training.

**Implementation.** Two distinct code bases are utilized. We implement our customized ZKP protocol in EMP-toolkit (Wang et al., 2016). EMP-toolkit is a C++ framework with efficient implementations of cryptographic building blocks. Our experiment is conducted on two Amazon Ec2 m1.xlarge machines (ARM machines), representing the prover and auditor. We report the average running time of 100 experiments for each different parameter. We adopt the Python implementation of Tramer & Boneh (2021) to train our models and assess accuracy and privacy. Recall that our protocol certifies that DP-SGD has been executed correctly. Then the DP guarantee is obtained by running a (public) privacy accounting technique which typically takes as input an arbitrary value of $\delta$ and produces a value for $\varepsilon$. We consider the standard the moments accountant technique (Abadi et al., 2016). In practice, a popular choice of $\delta$ is to set it as the reciprocal of the size of the dataset, which can be easily verified by the auditor in our framework after the prover has committed to the training dataset.

**Results.** Figure 1 shows the effectiveness and efficiency of *Confidential-DPproof* in training models for both CIFAR-10 and MNIST. We visualize the relationship between accuracy and running time and the relationship between certified privacy guarantees and running time. We run in total 225 training iterations and 55 training iterations for CIFAR-10 and MNIST, respectively, and reported the accuracy in the left column of Figure 1 and the certified DP guarantee ($\varepsilon$) in the right column of Figure 1, which both change as a function of training iterations. Figure 1 shows that the company can achieve 80% (respectively 88%) test accuracy with a certified privacy guarantee of $\varepsilon = 0.18$ (respectively $\varepsilon = 0.28$) in 4 hours (respectively 25 hours) when training a model on CIFAR-10. Institutions can achieve models with higher accuracy if they are willing to train for more iterations: as expected,

| #Attributes | Per-example running time | Per-minibatch running time | |
| --- | --- | --- | --- |
| | Gradient computation + clipping (s) | Model update (s) | Noise sampling (s) |
| 16 | 0.005 | 0.0003 | 0.0230 |
| 64 | 0.021 | 0.0011 | 0.0754 |
| 256 | 0.091 | 0.0098 | 0.2713 |
| 1024 | 0.324 | 0.0550 | 1.0873 |
| 4096 | 1.462 | 0.1614 | 4.5596 |
| 8192 | 3.068 | 0.3750 | 9.3851 |

Table 1: Efficiency and scalability of each building block of *Confidential-DPproof* in terms of running time for certifying DP guarantees of models trained on benchmark datasets for DP-SGD training. Obtaining a privacy guarantee in DP-SGD requires correct per-example gradient computation followed by clipping, carefully calibrated noise addition and correct model updates. ***Confidential-DPproof* efficiently proves each of these requirements of DP-SGD while protecting the confidentiality of training data, randomness and model parameters.** See Appendix C for experiments conducted on different machines.

the higher the accuracy, the higher the running time. For example, *Confidential-DPproof* enables the company to train a $91\%$ accurate model on CIFAR-10 with certified privacy guarantees of $\varepsilon = 0.55$ at the expense of 100 hours running time. Using the same amount of running time but using MNIST this time, *Confidential-DPproof* can train a model achieving $98\%$ test accuracy with a certified privacy guarantee of $\varepsilon = 0.74$. Note that DP-SGD training is costly even in the clear due to the need for per-example gradient computations instead of per-minibatch gradients (Subramani et al., 2021).

We further evaluate the scalability of *Confidential-DPproof* by analyzing the running time of the basic building blocks including per-example gradient computation, noise sampling and model updates as a function of the number of data attributes (values vary in $[16, 8192]$). The results are shown in Table 1 and averaged over 100 runs. The second column of Table 1 shows the effect of the number of attributes on the per-example gradient running time. *Confidential-DPproof* can prove the correct per-example gradient computation in a matter of seconds. For example, it takes less than 1 second to compute a per-example gradient when the number of attributes is 2048. The third and last columns of Table 1 report the running time of updating model updates and drawing normally distributed random variables at the end of each iteration for each minibatch. We comment that the per-example gradient computation running times scale linearly for multiple samples, while noise sampling and model updates running times scale linearly for multiple iterations.

## 5 RELATED WORK

**Data-driven privacy auditing.** The objective of privacy auditing is to assess the privacy guarantees offered by a given mechanism. This approach has been instrumental in evaluating the tightness (Nasr et al., 2021; 2023) and correctness (Tramer et al., 2022; Nasr et al., 2023) of DP machine learning algorithms. The concept of auditing DP originated from the hypothesis testing framework of differential privacy (Wasserman & Zhou, 2010). In this framework, an adversary aims to distinguish whether an output $O$ has been sampled from a mechanism run on either $D$ or one of its neighbouring datasets $D' = \{D \cup z\}$. The adversary's performance, measured by Type I (False Negative Rate, FNR) and Type II (False Positive Rate, FPR) errors, can be translated into a quantifiable privacy guarantee, provided these error rates are statistically valid. The conversion to a privacy guarantee can be done in any notion of differential privacy: Jagielski et al. (2020); Nasr et al. (2021) used $(\varepsilon, \delta)$-DP, Nasr et al. (2023) employed stronger assumptions and used Gaussian Differential Privacy (Dong et al., 2022), while Maddock et al. (2023) converted to Renyi Differential Privacy (Mironov, 2017).

The existing privacy auditing methodologies effectively assess system robustness and detect generic bugs, yet they fail to deliver formal proofs for asserted privacy guarantees. This poses a barrier for end-users intending to verify privacy claims. The obstacles to using existing privacy auditing techniques as privacy proofs are two-folded: 1) **Adversary Instantiation:** The verification of a privacy guarantee necessitates the creation of an adversary capable of executing provably optimal membership inference attacks *for each user* that would like to verify the privacy guarantees of the model. However, the design of such an adversary remains an unresolved challenge. Provable worst-case adversaries are only identified in restrictive scenarios (Nasr et al., 2021) or when additional assumptions are made on the threat model (Nasr et al., 2023; Maddock et al., 2023; Steinke et al.,

2023). Although there has been progress towards identifying universally worst-case adversaries for auditing (Andrew et al., 2023), the existing results are only empirical. The lack of a provably optimal adversary when performing privacy auditing calls for alternative strategies to verify the correctness of claimed privacy guarantees; 2) **Data Accessibility:** The formulation of a strong adversary (Shokri et al., 2017; Carlini et al., 2022) often implies access to the training data or a surrogate dataset with comparable attributes. Such access is often unrealistic in privacy-sensitive scenarios, highlighting the need for data-independent privacy verification algorithms – algorithms that can deliver reliable auditing without access to sensitive data. This will enable audits of privacy guarantees that are able to operate within the practical constraints of privacy-sensitive applications of ML. This sets the stage for our proposed methodology, which aims to address these challenges, employing the theory of Zero-Knowledge proofs to provide an efficiently verifiable proof for the users.

**Zero-knowledge proof, differential privacy and model training.** Prior works mostly use ZK (ZKProofs, 2022) to verify the correctness of ML inference (Weng et al., 2021b). Recently, it has been shown that cryptographic approaches are necessary for verifying the training of a model in a non-private setting (Zhang et al., 2022; Fang et al., 2023). There is only one work Shamsabadi et al. (2022) that developed ZKP protocols for training but they consider decision trees and the property being verified was fairness – rather than privacy. A handful of work has explored the combination of DP and ZKP. Narayan et al. (2015) change the language primitive of the Fuzz compiler such that the structure of the ZKP that proves the correctness of a response to a query does not leak any information about the dataset. Biswas & Cormode (2023) verify a specific DP histogram-like computation, while Sabater et al. (2022) use ZKP to verify decentralized DP averaging queries. In contrast, we propose and implement the first ZK protocol for verifying the correctness of DP training with DP-SGD. Cryptographically verifying properties of a training algorithm is always more challenging than verifying properties at inference time or verifying statistical queries on datasets; this is in large part due to the limited compatibility of cryptographic primitives with hardware accelerators. In our setting, this challenge is exacerbated because of the specificity of i) per-example gradient computations and clipping in DP training; and ii) the existence of randomness at different stages of DP training such as data subsampling to form minibatches. To address the challenges of efficiency and utility created by ZK proofs for DP training, we instantiate our ZK protocol from a vector-oblivious linear evaluation (Weng et al., 2021a), and build on SOTA interactive ZK protocols (Yang et al., 2021).

## 6 CONCLUSION

*Confidential-DPproof* can be used to confidentially verify the guarantees of a DP-SGD training run. We demonstrated the effectiveness and efficiency of *Confidential-DPproof* in enabling institutions to obtain a certificate for the DP guarantee they achieved while training high-accuracy models in a reasonable amount of time. *Confidential-DPproof* provides a cryptographic approach to auditing the protections afforded by DP-SGD and will thus enable trust from both end users and regulators.

The certificate provided by *Confidential-DPproof* upper bounds the information leakage from the *committed training data*. *Confidential-DPproof* reduces the incentive for a prover to manipulate data to achieve a desired output and then commit to the manipulated data. This is because the prover cannot control the randomness seed as it is been set through our interactive and private protocol with the auditor. However, one could further extend *Confidential-DPproof* by proving that no adversarial data manipulation, such as sharing information across data points, was performed prior to data commitment.

Our protocol is designed to attest the correctness of the key and common building blocks of DP algorithms including i) data subsampling; ii) gradient computation; iii) gradient clipping; iv) noise addition; and v) model update. Therefore, our protocol can support any randomized polynomial-time computable function and thus any (tractable) DP mechanism, as illustrated by our extension to DP-FTRL (Kairouz et al., 2021) in Appendix D. However, additional implementation in our ZK protocol is needed to support them. Once the correctness of the algorithm has been certified, the privacy guarantee can be computed and verified using any existing (public) privacy accounting technique. Therefore, *Confidential-DPproof* is also decoupled from the privacy accounting technique used. For example, our framework is compatible with recent advanced results for DP in the hidden state model (Ye & Shokri, 2022; Altschuler & Talwar, 2022) as our protocol keeps the entire trajectory (i.e., intermediate model updates) hidden from the auditor and can help to improve privacy by incorporating hidden states into the privacy accounting technique. Finally, our framework naturally benefits from future advances in cryptography, which would further improve the scalability of our approach.

## ACKNOWLEDGMENTS

Work of Gefei Tan and Xiao Wang is supported by DARPA under Contract No. HR001120C0087, NSF award #2310927, #2236819, and #2318974. Work of Tudor Cebere and Aurélien Bellet is supported by grant ANR-20-CE23-0015 (Project PRIDE) and the ANR 22-PECY-0002 IPOP (Interdisciplinary Project on Privacy) project of the Cybersecurity PEPR. Hamed Haddadi is supported by UKRI Open Plus Fellowship EP/W005271/1: Securing the Next Billion Consumer Devices on the Edge. Nicolas Papernot would like to acknowledge his sponsors, who support his research with financial and in-kind contributions: Amazon, Apple, CIFAR through the Canada CIFAR AI Chair, DARPA through the GARD project, Intel, Meta, NSERC through the Discovery Grant, the Ontario Early Researcher Award, and the Sloan Foundation. Resources used in preparing this research were provided, in part, by the Province of Ontario, the Government of Canada through CIFAR, and companies sponsoring the Vector Institute. Adrian Weller acknowledges support from a Turing AI Fellowship under EP/V025279/1, and the Leverhulme Trust via CFI. Ali Shahin Shamsabadi started this work while he was at The Alan Turing Institute. The views, opinions, and/or findings expressed are those of the author(s) and should not be interpreted as representing the official views or policies of the Department of Defense or the U.S. Government.

## ETHICS STATEMENT AND REPRODUCIBILITY STATEMENT

*Confidential-DPproof* enables interested third parties (e.g, users or government) to audit privacy claims made by model trainers (e.g., institutions), while protecting the intellectual properties of institutions. Therefore, *Confidential-DPproof* encourages the consideration of privacy while training machine learning models.

We have described our framework in details and our code is available at `https://github.com/brave-experiments/Confidential-DPproof` and `https://github.com/Gefei-Tan/Confidential-DPProof-zk`.

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

# A  DETAILS ON DATASET

Table 2 shows the details on our dataset.

| Dataset | Feature Extractor | #Attributes | MiniBatch size | Clipping norm | Noise multiplier |
|---------|-------------------|-------------|----------------|---------------|------------------|
| CIFAR-10 | ResNeXt | 4096 | 1024 | 0.1 | 3 |
| MNIST | ScatterNet | 3969 | 4096 | 0.1 | 3.32 |

Table 2: Datasets, feature extractors and hyperparameter values.

# B  MORE DETAILS ON THE CRYPTOGRAPHIC PRIMITIVES

To authenticate each input $x$ possessed by the prover, the prover and the auditor run a secure protocol with subfield Vector Oblivious Linear Evaluation functionality (Weng et al., 2021a) where:

- the prover obtains a uniform Message Authentication Code (MAC) $M_x$;
- the auditor obtains a global authentication key $\Delta$ and a uniform key $K_x$;

such that there is an algebraic relationship between them: $K_x = M_x \oplus x\Delta$.

Each step of the computation that modifies $x$, the prover and auditor will modify $K_x$ and $M_x$ in an agreed-upon way that will preserve this algebraic relationship given the new updated value of $x$. The auditor can detect if the prover made any modification to $x$ or/and did not perform the computation correctly only with the knowledge of $K_x$, not $x$ itself.

To further clarify the process, below we provide a simple summation example for which we: i) construct the corresponding circuit; iii) describe the circuit evaluation in detail; and iv) describe the verification of the correct behavior in detail.

Assume that the computation that we want to verify is the summation of $x$ and $x'$, $x'' = x + x'$.

**Circuit construction.** The prover and the auditor hold a circuit with 1 addition gate. A circuit is defined by a set of input wires along with a list of gates of the form $(\alpha, \beta, \gamma, T)$ where $\alpha, \beta$ are the indices of the input wires of the gate and $\gamma$ is the index of the output wire of the gate and $T$ is the type of the gate. The prover has $(x_\alpha, m_\alpha)$, $(x'_\beta, m_\beta)$ and $(x_\gamma, m_\gamma)$, and the verifier holds $K_\alpha, K_\beta, K_\gamma$ such that the above algebraic relationship hold between each pair of them. We denote such an authenticated value by $[\![x]\!]$ which means that the prover holds $(x_\alpha, m_\alpha)$ and the auditor holds $K_x$.

**Circuit evaluation.** Authenticated values are additive homomorphic, then they can locally compute $[\![x''_\gamma]\!] = [\![x_\alpha]\!] + [\![x_\beta]\!]$ as follows: i) the prover computing $x'' = x + x'$ and $M_{x''} = M_x + M_{x'}$; and ii) the auditor computing $K_{x''} = K_x + K_{x'}$.

**Verification.** Once the parties have authenticated values for the output of multiplication, the prover sends $M_{x''}$ to the auditor, and the auditor checks whether the algebraic relationship between $K_{x''}$, $x''$ and $M_{x''}$ holds to detect whether the prover did not do the computation correctly or the prover modified $x$ without informing the auditor.

# C  PERFORMING EXPERIMENTS ON DIFFERENT MACHINES

To demonstrate consistency across different hardware, we have included results from experiments conducted on different machines. We analyze the running time of the basic building blocks including per-example gradient computation, noise sampling and model updates as a function of the number of data attributes across three different machines with different CPUs: ARM, intel, and AMD (m1.xlarge, m7i.2xlarge, m7a.2xlarge, all with 16GB of RAM) in Table 3 and Table 4. These results demonstrate that the trends and conclusions drawn from our results remain consistent and reliable across different machines.

| #Attributes | Per-example running time | Per-minibatch running time | |
|---|---|---|---|
| | Gradient computation + clipping (s) | Model update (s) | Noise sampling (s) |
| 16 | 0.009 | 0.0004 | 0.0285 |
| 32 | 0.018 | 0.0014 | 0.0505 |
| 64 | 0.032 | 0.0020 | 0.1004 |
| 128 | 0.052 | 0.0051 | 0.1995 |
| 256 | 0.124 | 0.0062 | 0.3885 |
| 512 | 0.248 | 0.0187 | 0.7625 |
| 1024 | 0.533 | 0.0275 | 1.5874 |
| 2048 | 1.048 | 0.0689 | 3.1763 |
| 4096 | 2.121 | 0.1498 | 6.4705 |
| 8192 | 4.196 | 0.2844 | 12.657 |

Table 3: Efficiency and scalability of each building block of *Confidential-DPproof* in terms of running time for certifying differential private guarantees of models trained on real-world datasets. Experiment result on Intel machines.

| #Attributes | Per-example running time | Per-minibatch running time | |
|---|---|---|---|
| | Gradient computation + clipping (s) | Model update (s) | Noise sampling (s) |
| 16 | 0.012 | 0.0005 | 0.0344 |
| 32 | 0.025 | 0.0017 | 0.0676 |
| 64 | 0.047 | 0.0028 | 0.1320 |
| 128 | 0.095 | 0.0077 | 0.2587 |
| 256 | 0.188 | 0.0099 | 0.5352 |
| 512 | 0.375 | 0.0250 | 1.0065 |
| 1024 | 0.747 | 0.0467 | 2.0179 |
| 2048 | 1.489 | 0.0934 | 4.0886 |
| 4096 | 2.984 | 0.1944 | 8.0601 |
| 8192 | 5.944 | 0.3867 | 16.2481 |

Table 4: Efficiency and scalability of each building block of *Confidential-DPproof* in terms of running time for certifying differential private guarantees of models trained on real-world datasets. Experiment results on AMD machines.

## D  VERIFYING DIFFERENTIALLY PRIVATE FOLLOW-THE-REGULARIZED-LEADER (DP-FTRL)

In this section, we demonstrate that our protocol can support Differentially Private Follow-The-Regularized-Leader (DP-FTRL) (Kairouz et al., 2021). To do that, we demonstrate that implementing DP-FTRL using our ZK framework is simple.

**Implementation.** The DP-FTRL protocol only needs the following operations: arithmetic operations, public indexing of arrays, and sampling noise from the Normal distribution. All of these operations are already supported in our ZK protocol for DP-SGD. These operations used in the ZK proof only need to be rearranged/replicated to support the steps in DP-FTRL. In fact, the implementation of DP-FTRL in ZK is easier than the DP-SGD one as the DP-FTRL algorithm does not need shuffling and sampling which are all needed in DP-SGD. These two operations are complicated due to the existence of randomness.

**Results.** We train for one epoch over a dataset of size 100 with 16 attributes. We report the running time of verifying DP-FTRL and compare it with DP-SGD (for DP-SGD, we set the size of the minibatch to 5). Table 5 reports the running time of our framework for verifying DP-FTRL algorithm and its comparison to DP-SGD. The "Model update" column shows that the cost of tree aggregation of DP-FTRL is low. The "Noise sampling" column shows that noise sampling is the most expensive part of DP-FTRL which is expected as it needs to generate as many noise samples as the number of nodes in the tree. These results demonstrate that implementing a new DP algorithm using our framework is simple.

**Implementing other algorithms.** In general, an extension of our framework for verifying a new DP algorithm $\mathcal{A}$ can be seen as two separate steps:

| DP Algorithm | Gradient computation + clipping (s) | Model update (s) | Noise sampling (s) |
|---|---|---|---|
| DP-SGD | 0.822 | 0.084 | 0.519 |
| DP-FTRL | 0.824 | 0.161 | 3.460 |

Table 5: Verifying Differentially Private Follow-The-Regularized-Leader (DP-FTRL) and its comparison with DP-SGD.

- Using our unbiased randomness protocol to generate an unbiased randomness seed. If $\mathcal{A}$ requires randomness to be unbiased (for example for noise sampling and data subsampling), it can take advantage of our unbiased randomness protocol.

- Rewriting $\mathcal{A}$ in C++ using our ZK framework library. Our ZK framework is intuitive to use as it supports all essential floatpoint arithmetics. Our ZK framework will then automatically generate a circuit for $\mathcal{A}$ (step 3 of Algorithm 3). The input of the circuit and the public parameter can all be set using our framework. Once we have the circuit for $\mathcal{A}$, our framework will proceed to the rest of Algorithm 3.

