# OpenReview forum: "Confidential-DPproof: Confidential Proof of Differentially Private Training"
_ICLR.cc/2024/Conference — ICLR 2024 spotlight_

### Official Review · Reviewer_yrMg · 2023-10-20

**Soundness:** 4 excellent
**Presentation:** 3 good
**Contribution:** 4 excellent
**Rating:** 8
**Confidence:** 4

**Summary:**

The authors propose a zero-knowledge proof (ZKP) protocol, dubbed Confidential-DPproof, for an auditor to verify that a company (prover) has trained a ML model using DP-SGG at a certain privacy level, on a fixed private dataset (which should not be revealed to the auditor). Their method has three desirable properties:

1. An honest prover can convince an honest auditor that they have correctly implemented DP-SGD (and therefore that the resulting model is differentially private at a certain level, known to both parties);
2. A dishonest prover cannot convince an honest auditor that the trained model satisfies DP when it in fact does not; and
3. A dishonest auditor cannot bias the computations of an honest prover. In particular, a dishonest auditor cannot gain additional information about the training data, beyond what they would know from observing the output of a DP algorithm trained on the private data.

Experiments with Confidential-DPproof show that the ZKP mechanisms still allow for practically feasible runtimes for model training. The authors obtain strong model utility on CIFAR-10 and MNIST, while still enforcing strong DP guarantees ($\epsilon < 1$).

**Strengths:**

Confidential-DPproof provides a strong alternative to current methods for privacy auditing, which require instantiating membership inference adversaries to exploit the output of allegedly DP algorithms, thereby providing a lower bound on the privacy leakage. Unless the adversary can be shown to be optimal, however, this approach cannot provide an _upper_ bound on the privacy leakage. In general, implementing such attacks is also computationally difficult, and optimal adversaries are often intractable.

By approaching the problem from a different angle, the authors completely sidestep the need for optimal adversaries for verifying privacy guarantees. This is especially impressive because many of the strongest membership inference attacks require access to the private data in order to be trained (or at least a very good proxy), but this is unrealistic in practical scenarios requiring privacy. In contrast, their method does not require the auditor to have access to any private training data. As such, I believe this to be an exciting, non-incremental contribution, one which has the potential to change the paradigm for privacy auditing moving forward.

**Weaknesses:**

As ICLR is not primarily a security conference, it is likely that many readers will be unfamiliar with the terms and methodology used. As such, more discussion of the cryptographic primitives used would be helpful in improving the clarity of the paper. (See the "Questions" section below.)

Due to the additional computational overhead imposed by both DP-SGD itself, and the need to represent the steps of the algorithm in circuits which can be integrated with existing ZKP systems, the authors cannot train full neural networks. Instead, they rely on fixed feature extraction methods trained on public data or using other methods independent of the private training data, then train a logistic classifier on top of these representations. Even with these simplifications, gradient computation + clipping can still take over a second per sample in higher feature dimensions. However, as the authors point out in the related work, even the problem of computationally feasible non-private proof of learning is still unresolved. This would be an important avenue for future work in order for this auditing protocol to be applied to modern models with hundreds of billions of parameters trained on massive datasets.

**Questions:**

I do not have extensive background with ZKPs, so I would like to make sure my understanding of the paper is correct. Can the authors confirm if the following statements are true?

**Unbiased random seed generation:** From the honest prover's perspective, since $k$ was chosen uniformly at random and the auditor only knows $[[k]]$ (but nothing about $k$ itself), the random seed $s=k\oplus r$ is still uniformly random. From the honest auditor's perspective, since $r$ was chosen uniformly at random _after_ $k$ was fixed, $s$ must be uniformly random.

**Dataset commitment:** For verifying that the computations were performed on the committed dataset, we can think of it as follows. The data commitment is another key $K$ which depends on the dataset $\mathcal{D}$, but which gives the auditor no information about $\mathcal{D}$ (since it was an XOR with a private random quantity $M$ known only to the prover; this is similar to the relationship between $k$ and $[[k]]$ above). However, for each circuit $\mathcal{C}$ making up a step of the DP-SGD procedure, the prover can verify that the output of this step was computed on $\mathcal{D}$ using the agreed upon random seed, and this verification _only requires knowledge of $K$_, not $\mathcal{D}$ itself.

**DP-SGD privacy accounting:** This leads to my final question. It seems that the ZKP building blocks allow you to generate a proof for each iteration in DP-SGD, then the proof for the whole procedure is just the AND of all of these steps. In particular, this means that the auditor actually will see all of the intermediate models during the DP-SGD training procedure. There has been some recent work on improving the privacy guarantees of DP-SGD under the assumption that the algorithm output is only the _final_ model parameters $W^T$, rather than the entire trajectory [1]. Confidential-DPproof would be incompatible with this analysis, since there isn't a ZKP protocol (yet) which encodes the entire model training procedure, rather than just the individual steps.

Reference:
[1] Ye, Jiayuan, and Reza Shokri. "Differentially private learning needs hidden state (or much faster convergence)." Advances in Neural Information Processing Systems 35 (2022): 703-715.

---

> ### Author Response · Authors · 2023-11-16
> **Authors' response to reviewer yrMg (1)**
>
> We thank you for your careful reading, detailed and kind comments!
>
>
> > **Unbiased random seed generation: From the honest prover's perspective, since $k$ was chosen uniformly at random and the auditor only knows $[[k]]$ (but nothing about $k$ itself), the random seed $s=k\oplus r$ is still uniformly random. From the honest auditor's perspective, since $r$ was chosen uniformly at random after $k$ was fixed, $s$ must be uniformly random.**
>
> We confirm that this is correct, none of the prover and the auditor can bias the randomness seed. For example, the prover cannot bias the randomness seed $s$ by changing $k$ because the prover does not know the value of $r$ which is chosen by the auditor and is part of the computation of $s$. Therefore, even if the prover manipulates the value of $k$, the prover still sees $s$ as a uniformly random variable. The auditor, similarly, cannot bias the randomness seed $s$ because the auditor does not see $k$.
>
>
> We revised the last part of Phase 2 near the top of page 5 which says:
>
> “Observe that $s$ is random and cannot be biased by the prover or auditor, which guarantees an unbiased source of randomness when subsampling data or noising clipped gradients.”
>
> to:
>
> “Note that $s$ is random and cannot be biased by the prover or auditor,  which guarantees an unbiased source of randomness when subsampling data or noising clipped gradients. This is because the prover and the auditor cannot see values generated by other parties which contribute to the computation of $s$.”
>
> > **Dataset commitment: For verifying that the computations were performed on the committed dataset, we can think of it as follows. The data commitment is another key $K$ which depends on the dataset $\mathcal{D}$, but which gives the auditor no information about $\mathcal{D}$ (since it was an XOR with a private random quantity $M$ known only to the prover; this is similar to the relationship between $k$ and $[[k]]$ above). However, for each circuit $\mathcal{C}$ making up a step of the DP-SGD procedure, the prover can verify that the output of this step was computed on $\mathcal{D}$ using the agreed upon random seed, and this verification only requires knowledge of $K$, not $\mathcal{D}$ itself.**
>
> We confirm that this is correct, the verification only requires the knowledge of the key (not the data).
> In particular, to authenticate each input $x$ possessed by the prover, the prover and the auditor run a secure protocol with subfield Vector Oblivious Linear Evaluation functionality where:
> the prover obtains a uniform Message Authentication Code (MAC) $M_x$,
> the auditor obtains a global authentication key $\Delta$ and a uniform key $K_x$,
> such that there is an algebraic relationship between them: $K_x = M_x \oplus x \Delta$.
>
> Each step of the computation that modifies $x$, the prover and auditor will modify $K_x$ and $M_x$ in an agreed-upon way that will preserve this algebraic relationship given the new updated value of $x$. The auditor can detect if the prover made any modification to $x$ or/and did not perform the computation correctly only with the knowledge of $K_x$, not $x$ itself.
>
> To further clarify the process, below we provide a simple summation example for which we:
> 1. Construct the corresponding circuit
> 2. Describe the circuit evaluation in detail
> 3. Describe the verification of the correct behavior in detail
>
> Assume that the computation that we want to verify is the summation of $x$ and $x’$, $x’’=x+x’$.
>
> **Circuit construction.** The prover and the auditor hold a circuit with 1 addition gate.
>
> A circuit is defined by a set of input wires along with a list of gates of the form $(\alpha,\beta,\gamma,T)$ where $\alpha,\beta$ are the indices of the input wires of the gate and $\gamma$ is the index of the output wire of the gate and $T$ is the type of the gate.
>
> The prover has $(x_{\alpha},M_{\alpha})$,  $(x_{\beta},M_{\beta})$ and $(x_{\gamma},M_{\gamma})$, and the verifier holds $K_{\alpha},K_{\beta},K_{\gamma}$ such that the above algebraic relationship holds between each pair of them. We denote the authenticated value by $[[x]]$ which means that the prover holds $(x_{\alpha},m_{\alpha})$ and the auditor holds $K_x$.
>
> **Circuit evaluation.**
> Authenticated values are additively homomorphic, then they can locally compute $[x_{\gamma}]=[x_{\alpha}]+[x_{\beta}]$ as follows:
> The prover computing $x’’=x+x’$ and $M_{x’’}=M_x + M_{x’}$
> The auditor computing $K_{x’’}=K_x+K_{x’}$
>
> **Verification.** Once the parties have authenticated values for the output of multiplication, the prover sends $M_{x’’}$ to the auditor, and the auditor checks whether the algebraic relationship between $K_{x’’}$, $x’’$ and $M_{x’’}$ holds to detect whether the prover did not do the computation correctly or the prover modified $x$ without informing the auditor.
>
> We added this discussion in Appendix A and referred to it in Section 3.

---

> > ### Author Response · Authors · 2023-11-16
> > **Authors' response to reviewer yrMg (2)**
> >
> > > **DP-SGD privacy accounting: This leads to my final question. It seems that the ZKP building blocks allow you to generate a proof for each iteration in DP-SGD, then the proof for the whole procedure is just the AND of all of these steps. In particular, this means that the auditor actually will see all of the intermediate models during the DP-SGD training procedure. There has been some recent work on improving the privacy guarantees of DP-SGD under the assumption that the algorithm output is only the final model parameters $W^T$, rather than the entire trajectory [1]. Confidential-DPproof would be incompatible with this analysis, since there isn't a ZKP protocol (yet) which encodes the entire model training procedure, rather than just the individual steps.**
> >
> > In our current ZKP protocol, the auditor (1) does not see the intermediate results before being DP; (2) does not see the intermediate results after being DP; (3) does not see even the final model as we represent the whole procedure (not just a single iteration) as one single circuit and we verify the computation by proving that the commitment of the circuit output is consistent with its input without revealing any information to the auditor rather than the proof itself (please see line 5 of algorithm 3).
> >
> > (1) is necessary to protect the privacy of training data with respect to the auditor.
> >
> > (2) and (3) hide the entire trajectory and even the final model from the auditor (i.e., achieving perfect confidentiality), making our framework, Confidential-DPproof, compatible with existing [1] and future advanced results of DP by design. This follows the \emph{zero-knowledge} property of our framework described in Section 2 which says “If the prover and auditor execute $\Pi$ to prove that $\mathcal{C}(w)=y$, even a malicious auditor (who can behave arbitrarily) learns no information about the training data, intermediate model updates and trained model other than what can be inferred from the fact that the model is trained with the DP-SGD algorithm with $(\varepsilon,\delta)$-DP guarantees on the training data.”
> >
> > In summary, our protocol allows an auditor to verify the correct execution of noisy SGD performed by a prover while keeping all intermediate updates and even the final model hidden from the auditor. Then, the privacy guarantee can be computed and verified using any existing (public) privacy accounting technique such as [1] that you suggested. There is no need for cryptography here: the prover simply reveals the claimed privacy guarantee and indicates the accounting technique used to obtain it, and the auditor (or anyone) can verify that it is correct in the clear.
> >
> > We added this in Section 6 as:
> >
> > ``Once the correctness of the algorithm has been certified, the privacy guarantee can be computed and verified using any existing (public) privacy accounting technique. Therefore, \name is also decoupled from the privacy accounting technique used. For example, our framework is compatible with recent advanced results for DP  in the hidden state model [Jiayuan & Shokri, NeurIPS 2022; Altschuler & Talwar, NeurIPS 2022] as our protocol keeps the entire trajectory (i.e., intermediate model updates) hidden from the auditor and can help to improve the privacy by adding hidden states into the privacy accounting technique.’’
> >
> >
> >
> > [Ye \& Shokri, NeurIPS 2022] Jiayuan Ye, and Reza Shokri. "Differentially private learning needs hidden state (or much faster convergence)." NeurIPS 2022
> >
> > [Altschuler & Talwar, NeurIPS 2022] Jason M. Altschuler, Kunal Talwar: “Privacy of Noisy Stochastic Gradient Descent: More Iterations without More Privacy Loss.” NeurIPS 2022
> >
> >
> >
> >
> > > **This would be an important avenue for future work in order for this auditing protocol to be applied to modern models with hundreds of billions of parameters trained on massive datasets.**
> >
> > We agree with the reviewer that an interesting future direction is to audit the privacy of complex models for example through advancing research in cryptography. Therefore, we added the following in Section 6:
> >
> > ``Finally, our framework naturally benefits from future advances in cryptography, so that extending to massive models and datasets would be an interesting future direction.”

---

> > > ### Comment · Reviewer_yrMg · 2023-11-17
> > >
> > > Thanks to the authors for their response. The additional explanations and details for cryptographic/ZKP primitives are very helpful and I believe they will make the paper accessible to a broader audience. I've read the other reviews and responses, and I believe that the authors have adequately addressed both my concerns and the concerns raised by the other reviewers.
> > >
> > > I do have one new concern regarding novelty. Specifically, the authors claim that their method is "the first zero-knowledge protocol which enables auditors to directly verify the exact ε of a DP-SGD training run, and thus providing a certificate of privacy." However, I recently became aware of two papers which seem to be doing something quite similar:
> > >
> > > Ari Biswas & Graham Cormode. Verifiable Differential Privacy. arXiv preprint (2023). https://arxiv.org/abs/2208.09011v2 [The authors cited an earlier version of this work in their paper.]
> > >
> > > Arjun Narayan, Ariel Feldman, Antonis Papadimitriou, & Andreas Haeberlen. Verifiable differential privacy. Proceedings of the Tenth European Conference on Computer Systems (2015). https://dl.acm.org/doi/abs/10.1145/2741948.2741978?casa_token=1f0C9DZJI7IAAAAA:mhLCptViWeuvLomYyUqxOM4f-voEvPs2F6IqwK1fBP3cZNBzu2dTGeWovEuHz49od9ayLEDKGeYr9SI
> > >
> > > The authors cite the Biswas paper when explaining why it would be unacceptable to allow the auditor to choose the algorithm's randomness. However, it seems to me that this paper is specifically *avoiding* such an approach. The Narayan paper (with the same title) is not cited.
> > >
> > > Can the authors comment on the novelty of their work in light of these papers?

---

> > > > ### Author Response · Authors · 2023-11-20
> > > > **Thank you Reviewer yrMg! + Addressed your new concern.**
> > > >
> > > > Thank you so much for your time reading our responses to your review and others’ reviews! We are glad to hear that we addressed comments made in the reviews.
> > > >
> > > > Please find below the comparison between our work and both the Biswas & Cormode and Narayan et. al. papers:
> > > > 1. Although all of these works use zero-knowledge proof, we address different problems: i) Narayan et. al. focuses on the Fuzz compiler designed to implement simple private statistical queries (counts and sums). They change the language primitive of Fuzz such that the structure of the zero knowledge proof that proves the correctness of a response to a query does not leak any information about the dataset; ii) Biswas & Cormode aim to verify a specific DP 1 dimensional counting query;  while iii) we verify a DP-SGD training run of ML models. This is why we claimed that we propose ``the first zero-knowledge protocol which enables auditors to directly verify the exact ε of a DP-SGD training run”. We are happy to revise this if you think the claim of novelty is not sufficiently specific.
> > > >
> > > > 2. Verifying DP-SGD training of a model is much more challenging than verifying DP histogram or sum queries due to 1) being iterative; 2) the existence of the randomness at different stages of training such as data subsampling to form minibatch; 3) the complexity of gradient computation and clipping.
> > > >
> > > > 3. Biswas & Cormode’s randomness protocol only supports simple randomness (a Binomial distribution constructed from Bernoulli random variables) limiting their applicability to only the Binomial mechanism. This mechanism is not used in state-of-the-art DP training literature because (i) it typically has poorer utility than the Gaussian mechanism, and (ii) it is inherently an approximate DP mechanism which is not compatible with tight composition guarantees based eg on Rényi or Concentrated DP [Canonne et al., NeurIPS2020]. Narayan et. al did a further simplification by assuming that a trusted party provides the randomness to the prover.
> > > >
> > > > [Canonne et al., NeurIPS2020] Clément L. Canonne, Gautam Kamath, Thomas Steinke. The Discrete Gaussian for Differential Privacy. NeurIPS 2020.
> > > >
> > > > We added these two works in the last paragraph of our related work discussion, which now reads:
> > > >
> > > > `` Zero-knowledge proof, differential privacy and model training. Prior works mostly use ZK (ZKProofs, 2022) to verify the correctness of inference (Weng et al., 2021b). Recently, it has been shown that cryptographic approaches are necessary for verifying the training of a model in a non-private setting (Zhang et al., 2022; Fang et al., 2023). There is only one work (Shamsabadi et al. (2022)) that developed ZKP protocols for training but they consider decision trees and the property being verified was fairness – rather than privacy. There are only two works that combine differential privacy and zero-knowledge proofs. Narayan et. al. (2015) change the language primitive of the Fuzz compiler such that the structure of the zero knowledge proof, which proves the correctness of a query does not leak any information about the dataset. Biswas & Cormode (2022) aim to verify a specific DP histogram-like generation. However, we propose and implement the first zero-knowledge protocol for verifying the correctness of DP training with DP-SGD. Cryptographically verifying properties of a training algorithm is always more challenging than verifying properties at inference-time and verifying statistical queries on datasets; this is in large part due to the limited compatibility of cryptographic primitives with hardware accelerators. In our setting, this challenge is exacerbated because of the specificity of i) per-example gradient computations and clipping in DP training; and ii) the existence of randomness at different stages of DP training such as data subsampling to form minibatches. To address the challenges of efficiency and utility created by ZK proofs for DP training, we instantiate our ZK protocol from a vector-oblivious linear evaluation (Weng et al., 2021a), and build on state-of-the-art interactive ZK protocols (Yang et al., 2021).''
> > > >
> > > > We cited ​​Biswas & Cormode paper to support that an adversarial auditor can bias the computations and blame the prover. However, we acknowledge that this is potentially confusing (thanks for your careful reading and pointing this out!), so we removed it since we added a more detailed discussion about this paper in the related work section.

---

> > > > > ### Comment · Reviewer_yrMg · 2023-11-20
> > > > >
> > > > > Thank you for the clarification. Part of the confusion was due to a misunderstanding on my part; I see now that the authors specified that theirs is the first work which provides a ZKP specifically for DP-SGD, rather than the first to propose a ZKP for verifying *any* DP algorithm. The discussion of the other related works is helpful and the novelty of this paper in light of them is clear, and I believe the added discussion will make this clearer to readers as well.

---

### Official Review · Reviewer_PSjV · 2023-10-31

**Soundness:** 2 fair
**Presentation:** 3 good
**Contribution:** 3 good
**Rating:** 6
**Confidence:** 3

**Summary:**

This work proposes a protocol for auditing DP-SGD. The approach is based on zero-knowledge proof and does not require the auditor to access the model and raw data.

**Strengths:**

1. The problem of verifying privacy claims of algorithms is very important practically.
2. The proposed protocol based on zero-knowledge proofs does not require the auditor to access model parameters, data, and intermediate updates.
3. The authors take into account malicious auditors and dishonest provers in various aspects of the protocol such as random seed generation and

**Weaknesses:**

1. The auditor needs to know many implementation details, e.g. clip threshold, and number of iterations. Also, the protocol is specifically designed for DP-SGD. It seems that we need to design different protocols for different algorithms, even if we only make minor adjustments to the algorithm.
2. Many steps in the DP-SGD algorithm need to be proved in Phase 3. If one step is missing, for example, the auditor forgets to let the prover verify step vi, the total number of iterations, how would it affect validity of the privacy audit claims made by the auditor?
3. The proposed cryptographic approach does not scale to large models trained with DP-SGD.
4. It seems to me that the protocol only attempts to verify that every step of the DP-SGD algorithm is executed correctly as claimed, and the certified privacy parameters are simply derived based on the verified $\sigma$ and subsampling level, which is an upper bound on the actual privacy guarantee (as stated in the conclusion). Thus, when the certified upper bound exceeds the claimed value, we do not know whether there is a privacy failure. Note that even with 100% correct execution of DP-SGD, privacy failure may still exist due to other issues like finite precision computation of floats [1]. Thus, verifying all steps are executed correctly is not sufficient to audit privacy claims, and a privacy lower bound should still be necessary.
5. How does the approach compare to the recent work of [2]?
6. I have questions regarding the experiment setup. See the question section.

[1] Widespread Underestimation of Sensitivity in Differentially Private Libraries and How to Fix It, S Casacuberta, M Shoemate, S Vadhan, C Wagaman, CCS 2022

[2] Privacy Auditing with One (1) Training Run, Thomas Steinke, Milad Nasr, Matthew Jagielski, https://arxiv.org/abs/2305.08846


Typos:
1. Page 5, line 10 (the description of phase 2): "Next, the auditor generates... and sends.. to the auditor". The second "auditor" should be "prover"?
2. Page 6, line 2: we first generates -> generate

**Questions:**

1. What are the hyper-parameters: $C, \sigma, T$ in your experiments, and what are the corresponding theoretical upper bounds on $\epsilon, \delta$?
2. Compared to the privacy upper bound provided for the chosen $\sigma$, how accurate are the certified privacy guarantees?
3. The results do not show certified level of $\delta$.
4. The running time may vary across different machines and may not be a consistent measure of computational cost.

## Update
Increase my score to 6 after the authors quickly implemented ZKP for DP-FTRL.

---

> ### Author Response · Authors · 2023-11-16
> **Authors' response to reviewer PSjV (1)**
>
> We thank you for your review!
>
> > **The auditor needs to know many implementation details, e.g. clip threshold, and number of iterations.**
>
> Yes, as written at the beginning of Section 3, the prover and auditor agree on the DP-SGD training algorithm and specific values for its hyperparameters. This is also the case in all privacy auditing approaches and does not impose any overhead. These hyperparameters would typically be set by the prover to achieve high utility and shared with the auditor to verify the claimed privacy guarantees $(\varepsilon,\delta)$.
>
> > **Also, the protocol is specifically designed for DP-SGD. It seems that we need to design different protocols for different algorithms, even if we only make minor adjustments to the algorithm.**
>
> Our protocol, coupled with the unbiased randomness sampling protocol we propose, can support any randomized polynomial-time computable function and thus any (tractable) DP mechanism. In particular, our protocol is designed to attest the correctness of the key and common building blocks of DP algorithms: 1) data subsampling; 2) gradient computation; 3) sensitivity analysis;  4) noise addition; and 5) model update. Therefore one does not need to design new protocols for a new DP algorithm; however, extra implementation in ZK is needed to support them.
> Note that our approach is also decoupled from the privacy accounting technique used. For example, our framework is compatible with recent advanced results for DP  in the hidden state model [Jiayuan & Shokri, NeurIPS 2022; Altschuler & Talwar, NeurIPS 2022] as our protocol keeps the entire trajectory (i.e., intermediate model updates) hidden from the auditor and can help to improve the privacy by adding hidden states into the privacy accounting technique.
>
> We added this discussion in Section 6.
>
> [Ye & Shokri, NeurIPS 2022] Jiayuan Ye, and Reza Shokri. "Differentially private learning needs hidden state (or much faster convergence)." NeurIPS 2022
>
> [Altschuler & Talwar, NeurIPS 2022] Jason M. Altschuler, Kunal Talwar: “Privacy of Noisy Stochastic Gradient Descent: More Iterations without More Privacy Loss.” NeurIPS 2022
>
> > **Many steps in the DP-SGD algorithm need to be proved in Phase 3. If one step is missing, for example, the auditor forgets to let the prover verify step vi, the total number of iterations, how would it affect validity of the privacy audit claims made by the auditor?**
>
> We would like to stress that rather than verifying each step separately, our protocol allows the prover to generate a single proof to show that every step in the DP-SGD is computed correctly. This is reflected in step 3 and 4 of Algorithm 3 where the DP-SGD algorithm is encoded as a public circuit. So if the prover skips a step in the algorithm, it will not be able to produce a valid proof and the auditor will be able to tell. This follows from the soundness property of the underlying ZKP framework.
> We clarified this at the beginning of Section 3 by saying:
>
> ``Prover and auditor run our zero-knowledge protocol and provide a certificate for the claimed privacy guarantee $(\varepsilon,\delta)$. We represent the whole procedure as one single public circuit, allowing the prover to generate one single proof demonstrating to the auditor that it correctly executed all of the steps of the DP-SGD algorithm (see Algorithm 1).”

---

> > ### Author Response · Authors · 2023-11-16
> > **Authors' response to reviewer PSjV (2)**
> >
> > > **The proposed cryptographic approach does not scale to large models trained with DP-SGD.**
> >
> > We proposed the first zero-knowledge protocol which enables auditors to directly verify the privacy guarantees of a DP-SGD training run, while protecting the confidentiality of both their model and training dataset.  We agree with you that our protocol, similar to any other cryptographic protocols, comes with some overhead but we want to emphasize that such overhead allows us to achieve strong confidentiality, which could not be achieved otherwise.
> >
> > To reduce this overhead, instead of verifying the training of complex models on raw data which calls for heavy cryptographic machinery, we proposed a co-design of a DP training algorithm with a customized zero-knowledge (ZK) protocol in which we incorporate feature extractors to enable DP-SGD training of simpler models, thus being more efficient. Note that it has been recently shown that public feature extractors enable simpler models to achieve accuracy-privacy tradeoffs that are competitive with the state-of-the-art training of a complex model on a raw dataset with DP-SGD. Therefore, training simpler models is desirable in DP literature due to 1) the computational overhead imposed by DP algorithms themselves and 2) the fact that the utility cost of DP-SGD increases with the number of parameters. For example, as shown in [Figure 2, Tramer and Boneh, ICLR2021], privately training a simple logistic regression on features obtained by ScatterNet gives much better accuracy-privacy tradeoffs than privately training a CNN on raw data or even on extracted features.
> >
> > Finally, the overhead of our approach can be further reduced as research in cryptography continues, which is out of the scope of this paper.
> >
> > [Tramer and Boneh, ICLR2021] Florian Tramer and Dan Boneh. Differentially private learning needs better features (or much more data). In The International Conference on Learning Representations (ICLR), 2021.
> >
> >
> >
> > > **How does the approach compare to the recent work of [2]? [2] Privacy Auditing with One (1) Training Run, Thomas Steinke, Milad Nasr, Matthew Jagielski, https://arxiv.org/abs/2305.08846**
> >
> > The work of [2] tries to improve the computational costs of post hoc privacy auditing techniques that we cited in our paper but it still suffers from several limitations inherent to post hoc privacy auditing (described in the introduction and related work of our paper): in particular 1) establishing only lower bound on the privacy loss; and 2) the model and some data must be shared with the auditor to get a tighter lower bound on the privacy loss. Due to (1), an auditor using this post hoc approach can only hope to disprove a false claim of privacy as they cannot provide a certificate of the privacy claim made by the prover.
> > Instead, we propose a zero-knowledge proof (ZKP) protocol to proactively generate a certificate of privacy during training while preserving the confidentiality of all information. As highlighted by reviewer yrMg our proposal is
> >   - ``a strong alternative to current methods for privacy auditing’’ which includes [2];
> >   - ``has the potential to change the paradigm for privacy auditing moving forward’’; and
> >   - ``completely sidestep the need for optimal adversaries for verifying privacy guarantees’’.
> >
> > Following your suggestion, we cited [2] in the related work of our paper.
> >
> > > **Typos:
> > Page 5, line 10 (the description of phase 2): "Next, the auditor generates... and sends.. to the auditor". The second "auditor" should be "prover"?
> > Page 6, line 2: we first generates -> generate**
> >
> > Thanks for spotting these typos, we fixed them following your suggestions.

---

> > > ### Author Response · Authors · 2023-11-16
> > > **Authors' response to reviewer PSjV (3)**
> > >
> > > > **The running time may vary across different machines and may not be a consistent measure of computational cost.**
> > >
> > > Following your suggestion, we evaluated the running time of our framework using different machines and demonstrated that the trends and conclusions drawn from our results remain consistent and reliable across different machines.
> > >
> > > In particular, we analyzed the running time of the basic building blocks including per-example gradient computation, noise sampling and model updates as a function of the number of data attributes across three different machines with different CPUs: ARM, intel, and AMD (m1.xlarge, m7i.2xlarge, m7a.2xlarge, all with 16GB of RAM). Results are shown in Table 3 and Table 4 in the Appendix B of our paper and below:
> > >
> > > | #Attributes | ARM Gradient computation + clipping (s) | INTEL Gradient computation + clipping (s) | AMD Gradient computation + clipping (s) | ARM Model update (s) | INTEL Model update (s) INTEL | AMD Model update (s) | ARM Noise sampling | INTEL Noise sampling | AMD Noise sampling |
> > > |-------------|-----------------------------------------|-------------------------------------------|-----------------------------------------|----------------------|------------------------------|----------------------|--------------------|----------------------|--------------------|
> > > | 16          | 0.005                                   | 0.009                                     | 0.012                                   | 0.0003               | 0.0004                       | 0.0005               | 0.0230              | 0.0285               | 0.0344             |
> > > | 32          | 0.012                                   | 0.018                                     | 0.025                                   | 0.0006               | 0.0014                       | 0.0017               | 0.0407              | 0.0505               | 0.0676             |
> > > | 64          | 0.021                                   | 0.032                                     | 0.047                                   | 0.0011               | 0.0020                       | 0.0028               | 0.0754              | 0.1004               | 0.1320             |
> > > | 128         | 0.051                                   | 0.052                                     | 0.095                                   | 0.0024               | 0.0051                       | 0.0077               | 0.1461              | 0.1995               | 0.2587             |
> > > | 256         | 0.091                                   | 0.124                                     | 0.188                                   | 0.0098               | 0.0062                       | 0.0099               | 0.2713              | 0.3885               | 0.5352             |
> > > | 512         | 0.182                                   | 0.248                                     | 0.375                                   | 0.0144               | 0.0187                       | 0.0250               | 0.5668              | 0.7625               | 1.0065             |
> > > | 1024        | 0.324                                   | 0.533                                     | 0.747                                   | 0.0550               | 0.0275                       | 0.0467               | 1.0873              | 1.5874               | 2.0179             |
> > > | 2048        | 0.719                                   | 1.048                                     | 1.489                                   | 0.0625               | 0.0689                       | 0.0934               | 2.1725              | 3.1763               | 4.0886             |
> > > | 4096        | 1.462                                   | 2.121                                     | 2.984                                   | 0.1614               | 0.1498                       | 0.1944               | 4.5596              | 6.4705               | 8.0601             |
> > > | 8192        | 3.068                                   | 4.196                                     | 5.944                                   | 0.3750               | 0.2844                       | 0.3867               | 9.3851              | 12.657               | 16.2481            |
> > >
> > >
> > > We uploaded our code as supplementary material and considered standard experimental choices to ensure reproducibility. We are happy to perform any other experiment that you might have in mind.

---

> > > > ### Author Response · Authors · 2023-11-16
> > > > **Authors' response to reviewer PSjV (4)**
> > > >
> > > > > **What are the hyper-parameters: $C, \sigma, T$ in your experiments, and what are the corresponding theoretical upper bounds on $\epsilon, \delta$?**
> > > >
> > > > We added the values of the clipping norm $C$ and the noise multiplier $\sigma$ in Table 1 following the hyperparameter search done by [Tramer and Boneh, ICLR2021]:
> > > >
> > > > |Dataset| Clipping norm $C$ | $\sigma$ |
> > > > |-------|--------|--------|
> > > > | MNIST | 0.1  | 3  |
> > > > | CIFAR-10 | 0.1 | 3.32 |
> > > >
> > > > We clarified the number of training iterations at the beginning of Section 4.2 as:
> > > >
> > > > “We run in total 225 training iterations and 55 training iterations for CIFAR-10 and MNIST, respectively, and reported the accuracy in the left column of Figure 1 and the certified DP guarantees ($\varepsilon$) in the right column of Figure 1, which both change as a function of training iterations.”
> > > >
> > > > We also specified the value of $\delta$ in the caption of Figure 1 as follows:
> > > >
> > > > “The running time is computed based on the number of DP training iterations for various DP budgets $(\varepsilon,\delta=10^{-5})$.”
> > > >
> > > > [Tramer and Boneh, ICLR2021] Florian Tramer and Dan Boneh. Differentially private learning needs better features (or much more data). In The International Conference on Learning Representations (ICLR), 2021.
> > > >
> > > >
> > > > > **It seems to me that the protocol only attempts to verify that every step of the DP-SGD algorithm is executed correctly as claimed, and the certified privacy parameters are simply derived based on the verified $\sigma$ and subsampling level, which is an upper bound on the actual privacy guarantee (as stated in the conclusion). Thus, when the certified upper bound exceeds the claimed value, we do not know whether there is a privacy failure. Compared to the privacy upper bound provided for the chosen $\sigma$, how accurate are the certified privacy guarantees?**
> > > >
> > > >
> > > > Thank you for the question. Your first point is correct. In differential privacy, the privacy guarantees are obtained using privacy accounting techniques that typically give an upper bound. This is the best we can have as it is usually infeasible to compute the exact privacy guarantees of an algorithm (except in very simple cases). Our protocol allows an auditor to verify the correct execution of DP-SGD performed by a prover while keeping all intermediate updates and even the final model hidden from the auditor. Once the correctness of the algorithm has been certified, the privacy guarantee can be computed and verified using any existing (public) privacy accounting technique. There is no need for cryptography here: the prover simply reveals the claimed privacy guarantee and indicates the accounting technique used to obtain it, and the auditor (or anyone) can verify that it is correct in the clear. As a consequence, to answer your last question, the certified privacy guarantees will exactly match the privacy upper bound given by privacy accounting.
> > > >
> > > > We have clarified this in Section 4 and 6.
> > > >
> > > > > **Note that even with 100% correct execution of DP-SGD, privacy failure may still exist due to other issues like finite precision computation of floats [1]. Thus, verifying all steps are executed correctly is not sufficient to audit privacy claims, and a privacy lower bound should still be necessary. [1] Widespread Underestimation of Sensitivity in Differentially Private Libraries and How to Fix It, S Casacuberta, M Shoemate, S Vadhan, C Wagaman, CCS 2022**
> > > >
> > > > This is not the problem that we are addressing in this work. Note that our ZKP protocol implementation supports floating point representations, making it appropriate for potential future extensions of DP-SGD that might tackle this issue.
> > > >
> > > >
> > > > > **The results do not show certified level of $\delta$.**
> > > >
> > > > We do certify an $(\varepsilon, \delta)$-DP guarantee. As explained above, once we have certified that the DP-SGD algorithm has been executed correctly, the DP guarantee is obtained by running a (public) privacy accounting technique which typically takes as input an arbitrary value of $\delta$ and produces a value for $\varepsilon$. In practice, a popular choice of $\delta$ is to set it as the reciprocal of the size of the dataset, which can be easily verified by the auditor in our framework after the prover has committed to the training dataset.
> > > >
> > > > For clarity, we added the value of $\delta$ of the certified guarantee in the caption of Figure 1 and the above discussion at the beginning of Section 4.

---

> ### Comment · Reviewer_PSjV · 2023-11-21
> **Thank you for your response**
>
> Thanks for your response. You have addressed some of my concerns such as the certified level of $\delta$, runtime on different machines, and that the proof relies on a single circuit rather than verifying each step separately. However I still have questions, and some of my questions might have been misunderstood so I would also like to clarify them here. Pretty much all of my concerns fall into two categories: I. the flexibility and applicability of the framework. II. whether the certified privacy level is actually valid and independent.
>
> ### Flexibility of the proposed approach
> As you have mentioned proving a different algorithm would require a new implementation of ZKP, I wonder how much work is needed? Say we want to verify DP-FTRL (Algorithm 1 in the arxiv version of [1]), how difficult it is to implement a ZKP for this "variant" of DP-SGD?
>
> The scalability of large models is also an issue. While I agree that recent advancement in self-supervised learning enables private training of much smaller models on top of pre-trained models, there are recent works, e.g. [2] that try to privately train large foundation models due to copyright and privacy concerns of "public" datasets on the internet. Therefore auditing large models is still very relevant in practice.
>
> You also mentioned that knowing the specific algorithm and hyperparameters is needed "in all privacy auditing approaches and does not impose any overhead". This might be true for "white-box" auditing. However, existing works in privacy auditing also consider the "black box" scenario where the auditor only has access to the model weights. See Algorithm 3 in [3].
>
> ### Questions on the privacy claims.
> The proposed approach only verifies that the DP-SGD algorithm is executed correctly at every step. My main concern is that this is may not sufficient to provide an independently certified privacy claim, and that is the main reason I brought up [4], not to ask you to address this finite precision issue but to show as an example that even if every step of the algorithm is perfectly and correctly executed, privacy failures can still exist due to other potentially unknown implementation issues. If a similar failure indeed occurs, then I believe that the ZKP protocol would still provide a "certified" privacy upper bound, but the actual privacy loss may be much higher.
>
> In principle, I believe the certified privacy level given by the auditor should ideally be independent of the privacy accounting method and/or the theoretical derivations of the privacy upper bound, but this work heavily relies on those things to provide a certified privacy guarantee. Apart from the subtle implementation issues mentioned above, another issue actually comes from incorrect proof of privacy guarantees. Let's say some of the most popular privacy accounting methods actually have some subtle mistakes in their proof resulting in a higher privacy loss than their theoretical claims, I don't think the algorithm in this work can actually detect it and provide a certified privacy guarantee close to the true privacy loss.
>
> ### Conclusion
> As such, while this work proposes an interesting application of zero-knowledge proof to verify differential privacy claims, I do not think it is a revolutionary work that completely reshapes the current landscape of privacy auditing. In my opinion, this work is slightly orthogonal to existing works on privacy auditing because it only verifies that the algorithm is implemented correctly as opposed to providing an independently certified privacy guarantee. There are technical merits of the proposed approach such as not requiring access to the model and data, and taking into account both malicious auditor and prover.
>
> I acknowledge that verifying the implementation can serve as an important primary step in verifying the privacy claims and has practical values. In that case, the applicability of the proposed framework to a wider class of DP algorithms would be paramount. I believe this submission would be much stronger if the authors could demonstrate or explain that extending the framework to other algorithms can be done with relatively small effort.
>
> [1] Peter Kairouz, Brendan McMahan, Shuang Song, Om Thakkar, Abhradeep Thakurta, Zheng Xu, Practical and Private (Deep) Learning Without Sampling or Shuffling https://arxiv.org/pdf/2103.00039.pdf
>
> [2] Yaodong Yu, Maziar Sanjabi, Yi Ma, Kamalika Chaudhuri, Chuan Guo, ViP: A Differentially Private Foundation Model for Computer Vision
>
> [3] Privacy Auditing with One (1) Training Run, Thomas Steinke, Milad Nasr, Matthew Jagielski, https://arxiv.org/abs/2305.08846
>
> [4] Widespread Underestimation of Sensitivity in Differentially Private Libraries and How to Fix It, S Casacuberta, M Shoemate, S Vadhan, C Wagaman, CCS 2022

---

> > ### Author Response · Authors · 2023-11-22
> > **Thank you! + DP-FTRL is implemented and results added.**
> >
> > Thank you so much for your time reading our responses, and detailed comments!
> >
> > > **You have addressed some of my concerns such as the certified level of $\delta$, runtime on different machines, and that the proof relies on a single circuit rather than verifying each step separately.**
> >
> > We are glad to hear that we addressed these comments made by you.
> >
> >
> >
> > > **I acknowledge that verifying the implementation can serve as an important primary step in verifying the privacy claims and has practical values. In that case, the applicability of the proposed framework to a wider class of DP algorithms would be paramount. I believe this submission would be much stronger if the authors could demonstrate or explain that extending the framework to other algorithms can be done with relatively small effort.**
> > > **As you have mentioned proving a different algorithm would require a new implementation of ZKP, I wonder how much work is needed? Say we want to verify DP-FTRL (Algorithm 1 in the arxiv version of [1]), how difficult it is to implement a ZKP for this "variant" of DP-SGD?**
> >
> > Following your suggestion, we
> > 1. implemented DP-FTRL using our ZK framework,
> > 2. demonstrated that verifying a new DP algorithm using our framework is very simple and can be done with small efforts (we did it over the last 24 hours after receiving your comment).
> > 3. explained why (2) is the case.
> >
> > **Implementation.** We trained for one epoch over a dataset of size 100 with 16 attributes. We report the running time of verifying DP-FTRL and compare it with DP-SGD (for DP-SGD, we set the size of the minibatch to 5). We uploaded our code for verifying both DP-FTRL and DP-SGD as supplementary material.
> >
> > **Demonstration.** Below, we report the running time of our framework for verifying DP algorithms.
> > | #DP Algorithm | Gradient computation + clipping (s) | Model Update (s) | Noise Sampling (s) |
> > |---------------|-------------------------------------|------------------|--------------------|
> > | DP-SGD        | 0.822                               | 0.084            | 0.519              |
> > | DP-FTRL       | 0.824                               | 0.161            | 3.460              |
> >
> > 1. The Model Update column shows that the cost of tree aggregation of DP-FTRL is low.
> > 2. The Noise sampling column shows that noise sampling is the most expensive part of DP-FTRL which is expected as it needs to generate as many noise samples as the number of nodes in the tree.
> >
> > **Explanation.** Implementing a new DP algorithm ($\mathcal{A}$) using our framework is very simple and can be seen as two separate steps:
> > 1. Using our unbiased randomness protocol to generate an unbiased randomness seed. If $\mathcal{A}$ requires randomness to be unbiased (for noise sampling, random shuffling, etc.), it can take advantage of our unbiased randomness protocol.
> > 2. Rewriting $\mathcal{A}$ in C++ using our ZK framework library. Our ZK framework is very intuitive to use as it supports all essential floatpoint arithmetics. Our ZK framework will then automatically generate a circuit for $\mathcal{A}$ (step 3 of Algorithm 3). The input of the circuit and the public parameter can all be set using our framework. Once we have the circuit for $\mathcal{A}$, our framework will proceed to the rest of Algorithm 3.
> >
> > The DP-FTRL protocol only needs the following operations: arithmetic operations, public indexing of arrays, and sampling noise from normal distribution. All of these operations are already supported in our ZK protocol for DP-SGD. In our uploaded code we have shown how these operations used in the ZK proof need to be rearranged/replicated to support the steps in DP-FTRL. In fact, the implementation of DP-FTRL in ZK is easier than the DP-SGD one as the DP-FTRL algorithm does not need complicated operations of shuffling and sampling which are all needed for DP-SGD.
> >
> > Also, we want to highlight another property of our protocol which supports being generic: our protocol is designed such that intermediate updates and the entire trajectory are hidden → making it compatible with recent advanced results for DP in the hidden state model.
> >
> > We hope this addresses your concerns and that you will consider increasing your score.

---

> > > ### Comment · Reviewer_PSjV · 2023-12-01
> > > **Read your response - sorry for the late reply**
> > >
> > > I have read your response and thank you for putting in the effort to implement ZKP for another DP algorithm. I believe the new result indeed demonstrates that the framework can be easily applied to other algorithms. The scalability to larger deep neural network models is still an issue, but there are a lot of DP algorithms that are not as computationally demanding, so they can potentially benefit from this framework as well. As a result, my views of this work have increased slightly.  Considering other technical benefits (not requiring access to model parameters and data, robust to malicious behaviors on both sides), I am raising my score to 6.
> > >
> > > However, I believe that providing a privacy audit independently of the technical parts of the algorithm is very important, and this is the main reason why I am not giving a higher score. The proposed framework should serve as an initial step to verify the correctness of algorithm implementation, rather than an independent privacy audit. The authors should make this more clear in the final version of the paper.

---

### Official Review · Reviewer_y3Ya · 2023-10-31

**Soundness:** 2 fair
**Presentation:** 3 good
**Contribution:** 2 fair
**Rating:** 5
**Confidence:** 2

**Summary:**

This work studies important problem in data privacy research, the privacy auditing problem. The main approach is a zero-knowledge proof protocol for differential private machine learning.

**Strengths:**

The paper seems well-written.

**Weaknesses:**

Due to my lack background of zero-knowledge proof, it's difficult to evaluate the contribution.

**Questions:**

1. How this framework to give guidance to correct DP-SGD implementation if the algorithm did not pass the privacy auditing?

---

> ### Author Response · Authors · 2023-11-16
> **Authors' response to reviewer y3Ya**
>
> > **Due to my lack background of zero-knowledge proof, it's difficult to evaluate the contribution.**
>
>
> Our contribution lies at the intersection of machine learning, differential privacy and cryptography, and thus leverages and combines concepts from these multiple fields. We understand that it is difficult for a single person to have good expertise in all three domains, but we have done our best to make the paper as accessible as possible, and for this purpose we have added more discussion of the cryptographic primitives and our protocol. For example, we
>
> 1. Explained the data commitment, computation and verification in our framework with more details and provided an example of a summation, circuit construction, evaluation and verification [see Appendix A]
> 2. Explained that our framework represents the whole DP-SGD training procedure (not just a single iteration) as one single public circuit, allowing the prover to generate a single proof [see Section 3]
> 3. Explained with more details that our framework hides intermediate updates and the entire trajectory from the auditor, thus making it compatible with recent techniques of hidden states in DP which improve further the privacy of the prover
>
> We can concisely summarize our contributions as follows:
> 1. Introduced desiderata for privacy auditing;
> 2. Proposed the first zero-knowledge protocol which enables auditors to directly verify the exact ε of a DP-SGD training run, and thus providing a certificate of privacy;
> 3. Protected the confidentiality of private training data, model parameters, and intermediate updates;
> 4. Designed and implemented a specialized ZK proof protocol to efficiently perform the DP training;
> 5. Submitted our code as supplementary material to facilitate the future work in this direction that we initiated.
>
> Finally, please see the comments of reviewer yrMG who confirmed our contributions and said:
>
>   - ``I believe this to be an exciting, non-incremental contribution, one which has the potential to change the paradigm for privacy auditing moving forward”;
>   - ``a strong alternative to current methods for privacy auditing”;
>   - ``By approaching the problem from a different angle, the authors completely sidestep the need for optimal adversaries for verifying privacy guarantees. This is especially impressive’’.
>
>
>
>
>
> >  **How this framework to give guidance to correct DP-SGD implementation if the algorithm did not pass the privacy auditing?**
>
>
> Thanks for the good suggestion. Our framework can be extended to support identifying the cause of failure (i.e., which part of DP-SGD was not computed correctly) by generating separate proofs for each step of the DP-SGD. This comes at additional costs such as proving the consistency of intermediate values between proofs of each step. Given these complexities, we believe that a careful and thorough exploration is needed and we leave this to future work.

---

### Author Response · Authors · 2023-11-16
**Authors' (general) response to all reviewers**

Dear reviewers,

We thank you for your thoughtful reviews and for noting the strengths of our paper, namely:
1. An exciting, impressive and non-incremental contribution
   - ``I believe this to be an exciting, non-incremental contribution, one which has the potential to change the paradigm for privacy auditing moving forward” [reviewer yrMG]
   - ``a strong alternative to current methods for privacy auditing” [reviewer yrMG]
   - ``By approaching the problem from a different angle, the authors completely sidestep the need for optimal adversaries for verifying privacy guarantees. This is especially impressive’’ [reviewer yrMG]

2. Not requiring the auditor to have access to any private training data, model parameters, and intermediate updates  [reviewer PSjV and reviewer yrMg]

3. Verifying the exact privacy guarantees as opposed to a lower bound estimate [reviewer yrMG]

4. Robustness to dishonest provers and malicious auditors:
    - ``take into account malicious auditors and dishonest provers in various aspects of the protocol such as random seed generation”   [reviewer PSjV]

5. Practical in obtaining high utility while still enforcing strong DP guarantees:
   - ``Experiments with Confidential-DPproof show that the ZKP mechanisms still allow for practically feasible runtimes for model training. The authors obtain strong model utility on CIFAR-10 and MNIST, while still enforcing strong DP guarantees ($\epsilon < 1$).” [reviewer yrMg]

6. Studying the important and practical problem of verifying privacy claims of algorithms,
   - ``The problem of verifying privacy claims of algorithms is very important practically”. [reviewer PSjV]

7. Well written paper [reviewer y3Ya]

We respond below to all questions raised and have updated our manuscript, showing changes in red color. We summarize the main changes here:

1. Performed experiments on different machines (an ARM, intel machine, and a AMD machine on AWS), reported running time in Table 3 and Table 4 and demonstrated the consistency of our conclusions across different machines in Appendix B [reviewer PSjV]

2. Clarified that our framework hides intermediate updates and entire trajectory from the auditor, thus making it compatible with recent techniques of hidden states in DP which improve further the privacy of the prover in Section 6 [reviewer yrMg]

3. Discussed that our protocol, coupled with the unbiased randomness sampling protocol we propose, can support any randomized polynomial-time computable function and thus any (tractable) DP mechanism in Section 6 [reviewer PSjV]

4. Clarified that our framework represents the whole procedure (not just a single iteration) as one single public circuit, allowing the prover to generate a single proof in Section 3 [reviewer PSjV and yrMg]

5. Clarified that none of the prover and the auditor can bias the randomness seed in Section 3 [reviewer yrMg]

6. Discussed and clarified that our framework certifies a couple $(\varepsilon, \delta)$ that exactly matches the privacy upper bound given by privacy accounting. [reviewer PSjV]

7. Explained the data commitment, computation and verification in our framework with more details and provided an example of a summation, circuit construction, evaluation and verification in Appendix A [reviewer yrMg]

8. Added the value of hyperparameters in Table 1 and clarified that we use the same hyperparameter values as those in Tramer and Boneh, ICLR2021 [reviewer PSjV]

9. Fixed typos [reviewer PSjV]


If you have any further questions, comments or concerns, we are available and glad to respond promptly.

Thank you,

Submission8068 Authors

---

### Meta-Review · Area_Chair_JUrt · 2023-12-11

**Metareview:**

This paper introduces a technique to generate a cryptographic certificate of privacy during training called DPproof. The certificate is based on zero-knowledge proofs (ZKP). The proposed method is accompanied by significant experimental evaluation. Overall, the paper is well-written, and the proposed technique is clear.

Reviewers agreed that Confidential DPproof is a valuable technique for ensuring the correctness of DP algorithms. The ZKP approach adds to the growing number of techniques for privacy auditing by, among other things, proving correctness in the implementation of DP-SGD. I also note that the authors were very responsive to reviewers comments.

Overall, this is an interesting paper that brings a compelling dimension to privacy auditing.

**Justification For Why Not Higher Score:**

The reviewers noted some limitations with the proposed technique, particularly the connection with other aspects of privacy auditing. Specifically, reviewers suggested that the method focuses solely on ensuring the accurate execution of the DP-SGD algorithm at each stage, which is only one component in verifying privacy claims. Privacy breaches can still arise even when the algorithm is flawlessly executed due to other implementation factors (e.g., numerical precision issues).

**Justification For Why Not Lower Score:**

The paper brings an interesting new angle to privacy auditing by applying ZKP to ensure the correctness of DP-SGD implementations.

---

### Decision · Program_Chairs · 2024-01-16

Accept (spotlight)